# Multifaced Role of Dual Herbal Principles Loaded-Lipid Nanocarriers in Providing High Therapeutic Efficacity

**DOI:** 10.3390/pharmaceutics13091511

**Published:** 2021-09-18

**Authors:** Ioana Lacatusu, Teodora Alexandra Iordache, Mirela Mihaila, Dan Eduard Mihaiescu, Anca Lucia Pop, Nicoleta Badea

**Affiliations:** 1Faculty of Applied Chemistry and Materials Science, University Politehnica of Bucharest, Polizu No 1, 011061 Bucharest, Romania; ioana.lacatusu@upb.ro (I.L.); teodora.manasia@stud.chimie.upb.ro (T.A.I.); dan.mihaiescu@upb.ro (D.E.M.); 2Virology Institute Stefan S. Nicolau, Romanian Academy, Mihai Bravu Street No 285, 030304 Bucharest, Romania; mirela.mihaila@virology.ro; 3Faculty of Pharmacy, “Carol Davila” University of Medicine and Pharmacy, 6 Traian Vuia Street, 020945 Bucharest, Romania; 4RD Center, AC Helcor, Victor Babes St., 430082 Baia Mare, Romania

**Keywords:** dual-herbal nanocarriers, *Glycyrrhiza glabra* extract, Diosgenin, phytochemicals, free radicals, TNF-α, IL-6 cytokines

## Abstract

Although many phytochemicals have been used in traditional medicine, there is a great need to refresh the health benefits and adjust the shortcomings of herbal medicine. In this research, two herbal principles (Diosgenin and *Glycyrrhiza glabra* extract) coopted in the Nanostructured Lipid Carriers have been developed for improving the most desirable properties of herbal medicine—antioxidant and anti-inflammatory actions. The contribution of phytochemicals, vegetable oils and of lipid matrices has been highlighted by comparative study of size, stability, entrapment efficiency, morphological characteristics, and thermal behavior. According to the in vitro MTS and RTCA results, the dual herbal-NLCs were no cytotoxic toward endothelial cells at concentrations between 25 and 100 µg/mL. A rapid release of *Glycyrrhiza glabra* and a motivated delay of Diosgenin was detected by the in vitro release experiments. Dual herbal-NLCs showed an elevated ability to annihilate long-life cationic radicals (ABTS^•+^) and short-life oxygenated radicals (an inhibition of 63.4% ABTS^•+^, while the ability to capture radical oxygen species reached 96%). The production of pro-inflammatory cytokines was significantly inhibited by the newly herbals-NLC (up to 97.9% inhibition of TNF-α and 62.5% for IL-6). The study may open a new pharmacotherapy horizon; it provides a comprehensive basis for the use of herbal-NLC in the treatment of inflammatory diseases.

## 1. Introduction

There is substantial evidence suggesting that bioactive phytocompounds play an important role in the development and progression of pharmaco-medicinal therapy. The unsatisfactory outcome of current synthetic drugs for treating several diseases has led to the consideration of alternative medicine. A lot of investigations regarding phytocompounds have been developed in recent years, since, in nature, there are many phytocompounds with a diversity of health attractive and beneficial properties [1,2,3]. The World Health Organization reported that over 80% of the world’s population relies on herbal medicines (phytomedicine) for a particular aspect of their primary health care needs [4,5].

The great public interest and adherence in exploring bioactive herbal compounds (phytocompounds) for therapeutic purposes is constantly growing and will be more stimulated in the future. This is mainly due to toxicity challenges of synthetic drugs. The use of synthetic drugs has caused greater concern due to their high cost, as well as a considerable number of side effects [6]. Therefore, the exploration of an accessible and readily available therapy, with minimal side effects and multiple biological effects, is an absolute priority for the improved management of chronic diseases that cause disability worldwide, despite the remarkable advances made in the therapeutic field.

Herbal drugs from traditional medicine have been used to treat health problems since ancient times, but their effectiveness depends on the bioavailability over a sustained period [7]. Administration of herbal drugs to patients using traditional methods, however, is not always the most efficacious. Most phytocompounds experience several shortcomings such as bioavailability problems (low solubility), barrier function of the gastrointestinal tract and intensive primary metabolism [8,9]. Nanotechnology constitutes a distinctive and useful approach to overcome phytocompounds’ physicochemical limitations. Hence, nanostructured delivery systems for herbal medicine can potentially enhance the biological activity. Among the various colloidal delivery systems, nanostructured lipid systems are a promising strategy to overcome the main constraints of bioactive compounds from various herbal sources which restrict their application [10,11,12].

Nanostructured lipid carriers (NLC) have many privileges such as increased surface area, high biocompatibility, increased drug solubility and storage stability, enhanced bioavailability, minimized side effects, prolonged half-life and efficient targeted drug delivery [13,14,15,16]. NLC consists of biocompatible and biodegradable lipids and surfactants, which are ideal for lipophilic and poorly soluble drugs in water, thus increasing their oral absorption. In addition, due to lymphatic transport, primary metabolism is reduced, along with an increase in bioavailability.

Despite the pharmacological relevance of oils and extracts from various herbal sources, the therapeutic potential of many herbal active principles is still insufficiently explored in the nanotechnology–health domain. Given the current global trends, the use of these herbal products (phytocompounds) to provide bioactive natural principles that manifest multiple therapeutic effects and diminished or even non-existent side effects opens new perspectives in the pharmacotherapy of various diseases. As a result, in the present study, we were interested in the development of lipid nanosystems rich in herbal bioactive principles (phytochemical mixtures selected from the category of lipophilic and hydrophilic principles) and the demonstration of their therapeutic efficacy. The two herbal principles of lipophilic and hydrophilic nature, coopted in the same nanostructured lipid delivery system are Diosgenin, *DSG* (natural lipophilic active from wild yam extract, *Dioscorea villosa*) and licorice extract (hydrophilic plant active from *Glycyrrhiza glabra*). The goal is associated with improving the bioavailability of both herbal extracts (by ensuring a sustained and constant release of the two bioactive actives), protecting them against gastrointestinal degradation (avoiding primary metabolism), while coupling both types of therapeutic actions, anti-inflammatory, and antioxidant effects. The association of herbal bioactive principles with NLC systems offers opportunities both related to the development of new biocompatible formulations with better properties and to increase the efficacy of delivered bioactive phytocompounds.

Diosgenin (DSG, 3β-hydroxy-5-spirosten) is a natural steroid saponin (Figure 1) that is found predominantly in wild yam (*Dioscorea villosa*), but also in fenugreek (*Trigonella foenum greaceum*) [4]. It presents a wide range of pharmacological activities highlighted in numerous preclinical and clinical studies. DSG manifests therapeutic effects in several types of cancer (including breast cancer, osteosarcoma, colon cancer, leukemia, and prostate cancer), has a protective role in cardiovascular diseases (such as thrombosis and atherosclerosis), relieves diabetes and hyperlipidemia, regulates disorders neurodegene-rative, increases cell proliferation in human skin models and shows marked effects in regulating menopausal symptoms [17,18].

Although DSG provides a real and broad basis for its use in pharmacotherapy, its clinical application in the mentioned therapies is limited due to undesirable characteristics, such as high lipophilicity, poor pharmacokinetic profile, relatively short half-life, low bioavailability, and instability under different physiological conditions [19]. Pharmacokinetic results showed an absolute low bioavailability of DSG in only 6–7% of rats [20,21], which is due to its low solubility and extended primary metabolism. Due to these shortcomings, the design of a suitable delivery system for DSG is a major problem.

Numerous approaches have been taken to improve the efficacy and bioavailability of diosgenin, including the preparation of DSG-nanocrystals [20], polymeric DSG nanoparticles [22,23], or by conjugation with hydrophilic polymers, for example with mPEG [24]. The enhancement of DSG bioavailability has also been demonstrated in the case of nano-DSG, using layers of chitosan or albumin [25] or liquid crystals with b-cyclodextrin inclusion complexes [21]. DSG-cyclodextrin complexes, DSG polymeric nanoparticles or DSG nanocrystals showed aqueous solubility and superior oral bioavailability compared to DSG. To our knowledge, no studies have been identified on DSG nanoformulation in nanostructured lipid carriers (NLC).

Licorice root extract is one of the most famous herbal medicines in the world, found in the dried roots of *Glycyrrhiza uralensis Fisch., Glycyrrhiza inflata Bat.* and *Glycyrrhiza glabra* L. The biological and therapeuic activity of *Glycyrrhiza glabra* (*GlyG*) has been extensively investigated in Asia and Europe [26]. Phytochemical studies have shown that *GlyG* contains a variety of phytochemicals, such as flavone glycosides, triterpenoid saponins (mainly Glycyrrhizin, 13%), polysaccharides, sitosterol and amino acids [27]. *GlyG* is widely used in traditional Chinese medicine, being recognized in Chinese pharmacopoeia to treat asthma, gastritis, hepatitis and toxicosis [28]. It is also noteworthy that *GlyG* can also supplement other drugs to increase efficacy and reduce toxicity [26].

*Glycyrrhizic acid* (*GA* or Glycyrrhizin) is the most important bioactive ingredient in licorice root, being the main triterpene glycoside in *GlyG. GA* is an amphiphilic molecule (Figure 1b) consisting of one molecule of 18-β-glycyrrhetinic acid (hydrophobic part) and two molecules of glucuronic acid (hydrophilic part). *GA* has a wide range of pharmacological activities (antiviral, anti-inflammatory, anticancer, hepatoprotective, etc.) and is commonly used to treat acute and chronic liver damage, viral hepatitis, hepatic steatosis, hepatic fibrosis, hepatoma, other viral myocarditis such as psoriasis or prostate cancer [28]. One of the explanations for the presence of multiple pharmacological activities of *GA* is the membrane modification activity. The influence of membrane permeability, fluidity, pore formation and trans-membrane potential changes by *GA* have been recently reported by Selyutina et al. [26]. However, *GA* is accompanied by a lack of reduced bioavailability after oral administration [28]. After ingestion of licorice, glycyrrhizic acid is subjected to metabolic processes, being hydrolyzed to glycyrrhetic acid by enzymes of the intestinal microflora [29]. To avoid *GA* degradation, *Glycyrrhiza glabra* has been incorporated into liposomes and hyalurosomes [30].

## 2. Materials and Methods

### 2.1. Materials

l-α-phosphatidylcholine and poly(ethylene glycol)-*block*-poly(propylene glycol)-*block*-poly(ethylene glycol)/Poloxamer 188 were purchased from Sigma Aldrich Chemie GmbH (Munich, Germany). Hydrogen peroxide, Dimethyl sulfoxide (DMSO) and Polysor bate 20 were acquired from Merck (Darmstadt, Germany). The vegetable oils—primrose oil (from *Oenothera biennis*) and soybean oil (from *Glicyne max*) were bought from Textron Plimon S.L.U. (Barcelona, Spain). Glycerol monostearate (GMS) and cetyl palmitate (CP) were purchased from Cognis GmbH (Monheim am Rhein, Germany) and Acros Organics (Morris Plains, NJ, USA), respectively. The herbal extracts—*Glycyrrhiza glabra* extract, *GlyG* (standardized in 10% *Glycyrrhizic acid*/Glycyrizin), and *Discorea villosa* extract (standardized in 95% Diosgenin, DSG)—were supplied by Organic Herb Inc. (Changsha, China). The characterization of *Glycyrrhiza glabra* extract by ESI-High Resolution-Mass Spectrometry can be found in the Appendix A [31,32]. Tris[hydroxymethyl] aminomethane, potassium persulfate, 5-Amino-2,3-dihydro-1,4-phthalazinedione (Luminol), 2,2-azinobis-(3-ethyl benzthiazoline-6-sulfonic acid) (ABTS), 6-hydroxy-2,5,7,8-tetramethyl chroman-2-carboxylic acid (Trolox) were obtained from Sigma Aldrich (Darmstadt, Germany).

Cell culture and treatments: The HUVEC cell line is a normal, endothelial, standardized line isolated from the vascular endothelium of an umbilical cord. It is an adherent line, and the cells have a paved appearance and were purchased from American Type Culture Collection (ATCC). Adherent cells were routinely maintained in culture in DMEM: F12 medium added by 2 mM L-glutamine, 10% fetal bovine serum, 100 units/mL penicillin, 100 μg/mL streptomycin (Sigma Aldrich, St. Louis, Mo, USA) and incubated at 37 °C in 5% CO_2_ humidified atmosphere. For cytotoxicity assessment, normal cells (10,000/100 μL) were cultured (in the presence of herbal-NLC) in 96-well flat-bottom plates (for MTS assays) or 16-well E-type plates (for RTCA assay) in DMEM/F12 supplemented with 10% fetal bovine serum (FCS) and 2 mM L-glutamine (24 h). Increasing concentrations of NLC were added; stock solutions were prepared by dissolving 20 mg of NLC in 1 mL of DMSO. Induction of the inflammatory effect in normal HUVEC cell cultures was performed after treatment with 50 μM H_2_O_2_ for 24 h. After 24 h, the adherent cells were treated with different concentrations of NLC for different periods of time. Cellular treatments were performed using concentrations of 400, 200, 100, 50, 25, 12.5, 6.25 and 3.125 μM free- and herbal loaded-NLC. Cells were detached with a nonenzymatic solution of PBS/1 mM EDTA, washed twice in PBS.

### 2.2. Preparation of Individual and Dual-Herbal Actives—Lipid Nanocarriers

The individual- and dual herbal loaded-NLCs were prepared by the melt-emulsification method followed by high pressure homogenization (HPH), according to the procedure previously described by our group [12].

Briefly, an aqueous phase consisting in a blend of emulsifiers (nominated in Table 1) and hydrophilic herbal extract—*GlyG* was fast added into a melted lipid phase (at 73 °C), formed by a mixture of solid lipids, vegetable oil (nominated in Table 1) and lipophilic herbal active *DSG*. The resulting emulsions were stirred for 15 min (73 °C), and subsequently subjected to high shear homogenization (High-Shear Homogenizer PRO250 type, Conecticut, USA) at 12,000 rpm for 1 min. Then, the pre-emulsions were processed by high pressure homogenization (APV 2000 Lab Homogenizer, Germany), using six homogenization cycles (196 s.) at 500 bar. The hot nanoemulsions were cooled at room temperature to allow the recrystallization of a lipid blend and the obtaining of individual-NLC (NLC-I/II-*GlyG*; NLC-I/II-DSG) and dual-herbal actives—lipid nanocarriers (NLC-I/II-DSG-*GlyG*). The solid-NLC formulations were obtained after a lyophilization step (0.08 mbar, −55 °C, 54 h) using an Alpha 1-2 LD Freeze Drying System Martin Christ (Osterode am Harz, Germany).

### 2.3. Characterisation Methods

#### 2.3.1. Particle Size Measurement

The analysis of average diameters (Zave) was performed by dynamic light scattering (DLS), using a Zetasizer ZS 90 (Malvern Instruments Inc., Worcestershire, UK), equipped with a solid-state laser (670 nm) at a scattering angle of 90° and a temperature of 25.0 ± 0.1 °C. Samples were diluted with double distilled water prior to DLS analysis to obtain a reliable light scattering signal. The particle size data were evaluated using intensity distribution. Each value of Zave and PdI was given as an average of three individual measurements.

#### 2.3.2. Zeta-Potential Measurements

Zeta potential characterizes the static stability of lipid nanoparticles, the counterion diffusion layer existing around each colloidal particle can hinder the aggregation, mainly through electrostatic repulsive force. The electrical characteristics of the NLC were measured using a microelectrophoresis instrument (Zetasizer Nano ZS, Malvern Instruments Inc., Worcestershire, UK). The electrical charge (zeta potential, *ξ*) of NLC was determined in a capillary cell; by using the Helmholtz–Smoluchowski equation (Equation (1)) the measured particle electrophoretic mobility was converted into zeta potential. Prior to analysis, NLCs were diluted 1:100 with deionized water and was adjusted with 0.9% NaCl solution to avoid multiple scattering effects and to reach a conductivity of 50 µS/cm, respectively:(1)$$\xi =EM\;\frac{4\pi \eta }{\varepsilon }$$

*ξ*—zeta potential, *EM*—electrophoretic mobility, *η*—viscosity of the dispersion medium and *ε*—dielectric constant.

#### 2.3.3. Morphological Evaluation of Herbal-NLC

To observe the morphological and structural properties of herbal-NLC, Scanning Transmission Electron Microscopy (Hitachi HD 2700 Scanning Transmission Electron Microscope) was used. To perform STEM analysis, the NLC aqueous dispersions were diluted in 20 mL of distillate water and deposited on standard Cu TEM grids with a carbon thin layer film. TEM images were obtained in bright field mode.

#### 2.3.4. Thermal Behavior of Lipid Bulk, Empty-NLC and Individual-and Dual Herbal-NLC

The melting and crystallization temperatures of bulk lipids and NLC were determined using differential scanning calorimetry (Netzsch DSC 204F1 Phoenix, Selb, Germany). Approximately 20 mg of lyophilized samples were weighed into aluminum pans; a closed empty pan was used as a reference. The samples were heated from 25 to 100 °C at 10 °C per minute under constant nitrogen flow (40 mL/min). After being held at this temperature, the NLC was then cooled from 100 to 25 °C. The DSC enthalpy-temperature profiles of the NLC were recorded throughout this procedure. The melting and crystallization temperatures, as well as the enthalpy changes associated with these transitions, were determined from the heating and cooling curves using the instrument software. The percent of crystallinity index (*CI* %) was calculated according to the Equation (2):(2)$$CI\;\left(\%\right)=\frac{\Delta H_{sample}}{\Delta H_{bulk\;lipid}\times C_{lipid\;phase}}$$
where Δ*H_sample_* and Δ*H_bulk lipid_* are the measured enthalpy of bulk lipid and sample/NLC, respectively. *C_lipid phase_* is the concentration of lipid phase in the NLC.

#### 2.3.5. Entrapment Efficiency Determination

The entrapment efficiency of lipophilic and hydrophilic herbal actives was determined by quantifying the Diosgenin (*DSG*) and Glycyrrhizic acid (*GlyG*) from NLC by HPLC analysis. The 0.05 g lyophilized NLCs were dispersed into 1 mL ethanol and the suspension was centrifuged for 15 min at 13.000 rpm (Sigma 2K15, Osterode am Harz Germany). The un-entrapped DSG and *GlyG* were determined by using a Jasco 2000 liquid chromatograph equipped with a UV detector at λ = 203 nm and Teknokroma column (25 × 0.46 cm). The mobile phase was composed by ACN:H_3_PO_4_ (0.5%), *v*/*v* (90:10) and the flow rate was 1 mL/min; the retention times were 7.36 min for Glycyrrhizic acid, and 13.54 min for DSG, respectively. The un-entrapped DSG and *GlyG* were determined using the calibration curve in the concentration range of 20–200 μg/mL for DSG (R^2^ = 0.9989) and 4–40 μg/mL for glycyrrhizic acid (R^2^ = 0.9989). The HPLC procedure for quantitative determination of DSG and Glycyrrhizin was validated for linearity, specificity, and accuracy [33,34,35,36]. The percent of *EE* was calculated using the Equation (3):(3)$$\mathrm{EE}\%=\frac{Initial\;amount\;of\left(DSG/GlyG\right)\;into\;NLCs-Amount\;of\;free\;\left(DSG/GlyG\right)}{Initial\;amount\;of\;\left(DSG/GlyG\right)\;into\;NLCs}\times 100$$

#### 2.3.6. The In Vitro Determination of Antioxidant Activity, by Chemiluminescence and ABTS Methods

The free-radical-scavenger activity of NLC was evaluated using two assays: chemiluminescence and ABTS methods. The chemiluminescence method uses a Chemiluminometer Turner Design TD 20/20 (Sunnyvale, CA, USA) and the chemiluminescence system composed of luminol (10^−5^ M)—H_2_O_2_ (10^−5^ M) and a Tris-HCl buffer solution, at pH = 8.6 [37]. The antioxidant activity of individual and dual herbal actives-NLC was calculated used the Equation (4) and compared with those of the native herbal principles, vegetable oils and free-NLC:(4)$$\mathrm{AA}\%=\frac{\left(I_{0}-I_{S}\right)}{I_{0}}\times 100$$

*I*_0_—the maximum chemiluminescence for the standards, at *t* = 5 s; *I_s_*—the maximum chemiluminescence for the sample, at *t* = 5 s.

In order to evaluate the ability of NLC to capture the long-life cation radicals, i.e., ABTS^●+^, a spectrophotpmetric ABTS assay was employed (UV-Vis-NIR Spectrophotometer, V670 Jasco, Tokio, Japan). ABTS^●+^ was produced by the reaction between 7 mM ABTS solution with 2.45 mM potassium persulfate (16 h, dark condition). All determinations were achieved in triplicate after 4 min against a blank sample prepared with 3 mL ABTS solution normalized to an absorbance of 0.700 (±0.02) at 734 nm and 2 mL of ethanol. The antioxidant activity of NLC was determined by treatment of 1 mL NLC (prepared by dissolving 0.05 g NLC in 25 mL of ethanol) with 3mL ABTS aq. and 1 mL EtOH. The inhibition of ABTS^●+^ (%) was calculated as inhibition using the Equation (5):(5)$$\%Inhibition\;{\mathrm{ABTS}}^{●+}=\frac{A_{0}-A_{s}}{A_{0}}\times 100$$
*A*_0_—the absorbance of the blank (unscavenged radical cation solution), while *A*_s_—the absorbance after the addition of the sample (free-NLC/loaded-NLC/native extract/vegetable oil).

#### 2.3.7. Phytocompounds (Diosgenin and Glycyrrhizic Acid) In Vitro Release

The dialysis bag diffusion technique was employed to evaluate the rate of DSG and *Glycyrrhizic acid* release from NLC prepared with primrose and soybean oils. In brief, 5 mL of each herbal loaded-NLC was placed into a dialysis bag (MWCO 12–14 kDa) and immersed into 50 mL of receiving medium (ethanol: phosphate-buffered solution, pH 6.4 = 70:30). The release environment temperature was kept at 37 °C and 300 rpm. At the predetermined time interval, 1mL of the samples was withdrawn and replaced with a fresh buffer solution. The amounts of released Diosgenin and *Glycyrrhizic acid* were measured by HPLC analysis considering the previously described entrapment efficiency method (Section 2.3.5).

#### 2.3.8. Cytotoxicity Assay (MTS)

To evaluate the cytotoxicity of NLC, a colorimetric method based on the MTS method was used. MTS is a yellow tetrazolium salt [3-(4,5-dimethylthiazol-2-yl)-5-(3-carboxymethoxyphenyl)-2-(4-sulfophenyl)-2H-tetrazolium] which can be reduced by the active metabolic cells, e.g., from yellow to colored formazans. In vitro experiments involved the use of 96-well plates with flat bottom (Thermo Scientific, Waltham, MA, USA) and the CellTiter 96 Aqueous One Solution Cell Proliferation Assay (Promega, Madison, WI, USA). Then, 1 × 10^4^ cells/well were cultured in 100 μL for 24 h, supernatant cultures were discarded, then cells were treated with different concentrations of NLC-DSG-*GlyG*, for 24 or 48 h. After incubation, 20 μL staining solution (MTS and PES), were added to each well and then the plates were incubated for 4 h at 37 °C.

The reduction of MTS to formazan was measured spectrophotometrically at λ = 492 nm, using a Dynex plate reader (DYNEX Technologies MRS, Chantilly, VA, USA). The cell viability was calculated with the Equation (6):(6)$$\mathrm{Cell}\;\mathrm{viability}\;\left(\%\right)=\frac{\left({\mathrm{Abs}}_{\mathrm{treated}\;\mathrm{cells}}-{\mathrm{Abs}}_{\mathrm{culture}\;\mathrm{medium}}\right)}{\left({\mathrm{Abs}}_{\mathrm{untreated}\;\mathrm{cells}}-{\mathrm{Abs}}_{\mathrm{culture}\;\mathrm{medium}}\right)}\times 100$$

Cell viability data were expressed as the mean values ± standard deviations (SD) of the experiments. Data were obtained in triplicate (*n* = 3) averaged and expressed as mean ± standard deviation (SD).

#### 2.3.9. Real-Time Cell Analysis (RTCA)

xCELLigence technology and RTCA-DP analyzer have been employed for in vitro quantification of viability versus cytotoxicity of developed herbal-NLC. 1 × 10^4^ HUVEC normal cells were seeded in 100 μL culture medium in each well (E-Plates 16, ACEA Biosciences, San Diego, CA, USA) and afterward were put in the xCELLigence DP-System and placed in a humidified incubator (5% CO_2_). The RTCA curves were automatically recorded on the xCELLigence System in real time. Cell impedance was measured (by using electronic sensors) and the small changes in impedance were continuously measured by RTCA-DP analyzer and after integration were expressed over time as Cell Index, *CI* (using RTCA Software 1.1, Mannheim, Germany). When cell proliferation reached a *CI* value over 1.0, scalar concentrations of NLC were added, and live cells were monitored.

#### 2.3.10. The Assessment of the Anti-Inflammatory Activity (ELISA Assay)

ELISA assay was performed on a “Ready-Set-Go” Affimetrix eBioscience Lab system. For determination of the pro-inflammatory IL-6 and TNF-α cytokine level, ELISA kits have been employed. The coating step consisted of overnight incubation at 4 °C of 96-well plates with specific capture antibodies. After a step of washing with PBS-Tween 20 0.05%, the standards and herbal-NLC were added and incubated overnight at 4 °C. By 1:2 serial dilutions, starting from 500 pg/mL for IL-6 and 1000 pg/mL for TNF-, standard curves of 8 concentration points were achieved. Incubation with specific detection antibodies for 1 h at room temperature was performed after a pre-wash with PBS-Tween 20. To detect target molecules, the Avidin-HRP enzyme was added for 30 min at room temperature, and then TMB substrate solution for 15 min. The NH2SO4 solution was used to stop the reaction and the optical densities were measured spectrophotometrically at 450 nm. The secretion of proinflammatory cytokines IL-6 and TNF-alpha was evaluated in HUVEC cell culture supernatants with or without induced inflammation at the end of treatment with compounds and/or H_2_O_2_. The conditioned media were protected, centrifuged for 30 min at 400× *g*, and the supernatants were stored at −80 °C until ELISA was performed. Values were obtained from standard curves performed for each cytokine and are expressed in pg/mL. Normalization of the results was done by representing the concentrations of cytokines released from the treatments performed against the untreated cell concentrations, which was 100%.

#### 2.3.11. Statistical Analysis

Data in the figures are expressed as mean value ± standard deviation (SD) for three measurements. Data analyses were performed using GraphPad Prism 7 (GraphPad Software Inc., La Jolla, CA, USA). The differences between the treatment and control groups were statistically analyzed using unpaired two tailed *t*-test and one-way ANOVA. Statistical significance was considered at *p* < 0.05 and insignificant statistically at *p* > 0.05 (NS).

## 3. Results and Discussion

### 3.1. Size, Morphologic Aspects and Physical Stability Features

The preliminary characterization of lipid nanocarriers was represented by the comparative determination of the average diameters of NLC that present a single incorporated plant principle, referred as NLC-individual (NLC-I/II-DSG and NLC-I/II-*GlyG*) and those that co-encapsulate both plant principles, hydrophilic and lipophilic, referred as NLC-dual (NLC-I/II-DSG-*GlyG*). According to the dynamic light scattering analysis, the lipid nanocarriers showed average diameters less than 200 nm, with a relatively narrow distribution of the lipid particle population, mostly (with small exceptions), the polydispersity indices between 0.17 and 0.25 (Figure 2). A comparative analysis of Zave characteristic of the two categories of NLC (prepared with oil from *Oenothera biennis* and *Glicyne max*, respectively), shows insignificant differences between them, both vegetable oils being suitable for obtaining lipid nanocarriers that have desired encapsulation characteristics for entrapping both mixtures of bioactive phytochemicals principles, DSG and *GlyG*.

Interesting results have been reported when increasing the amount of encapsulated licorice extract. A higher amount of encapsulated *GlyG* led to obtaining nanocarriers with a significantly smaller size compared to its analogue that encapsulates a lower amount of *GlyG* (e.g., for NLC-I-DSG-*GlyG* 1, Zave = 188.4 ± 0.764 vs. 122 ± 0.791 nm, for NLC-I-DSG-*GlyG* 4). This decrease in average diameters can be correlated with the potential surfactant role of glycyrrhizic acid from licorice extract [26,35], which complements the surface properties of Tween 20 and phosphatidylcholine. In addition, obtaining polydispersity indices < 0.2 (e.g., PdI of 0.183 ± 0.008 for NLC-I-DSG-*GlyG* 4 and of 0.225 ± 0.029, respectively for NLC-II-DSG-*GlyG* 4) reinforce the observation of obtaining relatively monodisperse systems, with a narrow size distribution.

The size of the lipid nanospheres loaded with DSG and *GlyG*, as well as aspects related to their morphology were provided by visualizing the lipid nanocarriers in the ZC phase contrast micrographs (Figure 3). Depending on the type of plant oil used in the preparation of lipid nanocarriers, some differences have been observed. For example, if in the case of nanocarriers prepared with soybean oil (NLC-II systems), the presence of spherical nanoparticles with diameters ranging from 50 to 180 nm was observed (Figure 3a) and no other internal structural shapes were detected, in the case of NLC-I systems, small nanospheric inclusions < 5 nm could be easily observed (Figure 3b). These inclusions of the nanoparticles uniformly dispersed in the lipid core can be attributed to the existence of the active plant principles captured in the different nanocompartments created by the plant oil or in the imperfections generated by the hydrocarbon chains of the fatty acids in the solid lipid mixture.

The evaluation of the physical stability of NLC was determined by measuring the electrokinetic potential, based on electrophoretic mobility. The obtained results highlighted that NLC presents appropriate physical stability, with zeta potential values between −39 mV and −56 mV for NLC prepared with plant oil from *Oenothera biennis* and −35 mV and −48 mV for NLC prepared with soybean oil (Figure 4). Highly electronegative values of the electrokinetic potential demonstrate the existence of repulsion phenomena between lipid particles in aqueous suspension, favorable for preventing the aggregation of lipid nano-carriers loaded with both lipophilic and hydrophilic plant principles.

The observation of interest that emerges from the zeta potential results refers to the presence of denser populations of electronegative charges in the case of NLC-I systems, compared to NLC-II systems (Figure 4). Of note, there is a significant change in electrokinetic potential values, which occurs as the *GlyG* concentration increases, for example ξ = −48.7 mV (for NLC-II-DSG-*Gly* 1) vs. −37.7 mV (for NLC-II-DSG-*Gly* 4). However, the values of the electrokinetic potential remain strongly electronegative which suggests an adequate stability over time of all the prepared NLC-I and II, respectively. These results lead to the conclusion that the physical stability of the prepared NLC is influenced by the percent of hydrophilic plant principle, which most likely remains attached to the outside of the lipid core/in the surfactant coating and actively participates in the modification and redistribution of surface loads.

### 3.2. Evaluation of Structural Changes of Lipid Nanocarriers, after the Simultaneous Co-Optation of Both Categories of Active Plant Principles

In addition to obtaining the desired size and physical stability, the development of lipid nanocarriers with superior qualities also involves an amorphous state of the lipid core suitable for the sustained release of the two bioactive principles present in the same delivery nanostructured carrier. The study of the thermal behavior of NLC loaded with different active principles was performed in parallel with free nanocarrier systems (NLC-I/II) and with the two physical mixtures of solid lipids and plant oils. The melting points of the lipids blend present in the two physical mixtures appeared as endothermic peaks at approx. 48 °C, 55 °C and 62 °C, respectively (Figure 5). The development of the three endothermic peaks in the physical mixtures cumulated with the easily noticeable changes of melting points (in pure lipids) suggests the complexity of the lipid matrix as well as the influence of unsaturated fatty acids from the plant oils. These endothermic peaks were found in lipids brought to the nanometer scale/NLC formulations, in the form of a wider melting range of 42 to 60 °C. The peak broadening is caused by a more complex and disordered network structure. The flattening degradation of the onset temperature may also indicate that a maximum solubility of the lipophilic active principles (DSG) in the lipid mixture is achieved. It is known that active lipophilic principles have appreciable solubility in melted lipids.

Minor endotherms occurring at lower temperatures (39–43 °C) may be due to melting of the vegetable oil (and/or co-surfactant) domains, followed by major endotherms corresponding to melting of the major lipids in the mixture. The melting of lipids in NLC took place at a slightly lower temperature compared to pure lipids, due to the smaller particle size. Similar results of the decrease of the melting point of lipids due to the size (after bringing them to the nanometric scale) are widely reported in the literature [38]. This can be explained according to classical thermodynamics, in which the melting point of the solid solution is expected to decrease due to the increased entropy of the blend [39]. Additionally, the higher ratio between the specific surface and the particle volume of smaller size, compared to the raw material (phenomenon described by the Gibbs—Thompson effect), results in a decrease in the melting point.

In both nanocarriers, NLC-I/II-DSG and NLC-I/II-DSG-*GlyG*, there is an obvious widening of the melting range compared to NLC-I/II (free nanocarriers), which suggests a significant disruption of the network formed by the various structural lipids, because of accommodating the herbal active principles. A slightly sharper endothermic allure was highlighted for NLC-I/II-*GlyG*, which confirms a higher presence of hydrophilic *GlyG* in the surfactant coating, and implicitly an insignificant disturbance of the lipid core (Figure 5). The most pronounced flattening of DSC endotherms was recorded in the case of dual systems that co-encapsulate both active principles, DSG and *GlyG*.

With respect to the lipid crystallinity of nanostructured carriers (data summarized in Table 2), a more advanced degree of crystallinity was detected in NLC loaded with DSG and a mixture of DSG and *GlyG*, compared to NLC-free.

The high crystallinity of NLC-DSG and NLC-*GlyG* is clearly evidenced by the increase in ΔH values compared to NLC-free (e.g., ΔH = 70.9 and 60.6 J/g for NLC-I-DSG/NLC-I-*GlyG* vs. 52.2 J/g for NLC-I). This decrease in melting enthalpy is accompanied by an increase in crystallinity, from 28.3% in NLC-I to 38.4% and 32.8% for NLC-I-DSG and NLC-I-*GlyG*, respectively. Interestingly, in NLC-dual systems, crystallinity remains at a value close to that of NLC without encapsulated active ingredient (e.g., CI of 29.7% for NLC-I-DSG-*GlyG* vs. 28.3% for NLC-I and a crystallinity degree of 36% for NLC-II-DSG-*GlyG* versus 34.6% for NLC-II). These results may be an indication that maximum solubility of DSG has been achieved in the lipid mixture or in the plant oil nanocompartments.

### 3.3. Encapsulation Efficiency of Individual and Dual Phytochemicals Captured into Nanostructured Carriers

The quantitative results highlighted the remarkable ability of NLC-I/II to capture both categories of phytochemicals with different polarities, by means of preferential affinities for the lipophilic lipid core (Diosgenin) or the coating made by the mixture of surfactant and co-surfactants (case of *Glycyrrhiza glabra* extract). The strong hydrophilic character of the *GlyG* did not constitute an impediment for the capture with almost maximum encapsulation yields of the main component of hydrophilic *GlyG*, respectively of glycyrrhizic acid (the determined EE were 96.5% ± 0.57 and 95.8% ± 0.54, for NLC-I/II-*GlyG*). In contrast, in the case of dual-phytochemicals NLC, which co-encapsulates *GlyG* and DSG, as the amount of encapsulated *GlyG* increased from 0.5 to 2%, the encapsulation efficiency suffered slight decreases (Figure 6), but they remained in a fairly high range (i.e., 87.6% to 85.4% for NLC-I). Similar behavior was encountered for NLC systems prepared with soybean oil, EE being in this case between 83 and 89.5%.

Regarding the lipophilic phytochemicals/DSG, its higher affinity for the lipid mixture formed by the two solid lipids with primrose oil can be noticed (89.5% ± 2.55 DSG for NLC-I-DSG vs. = 85.7% ± 3.38 for NLC-II-DSG). A potential explanation in this case can be attributed to the better solubility of DSG in primrose oil which has a lipid profile rich in unsaturated fatty acids, the most abundant being linoleic acid (~74%), γ-linolenic acid (9%) and oleic acid (7%) [40]. Usually, the solubility of an active ingredient is lower in solid lipids than in oils and this would be mainly due to the presence of the long hydrocarbon tails in solid fats. Due to the structural differences between solid and liquid lipids, lipid molecules cannot be arranged in a perfect crystalline structure during solidification. Thus, the uniformity of the crystal structure in the lipid matrix of NLC is disrupted. As such, the presence of imperfections in the lipid network is directly responsible for high values as well as for the improvement of the encapsulation efficiency.

Concerning the dual phytochemicals-nanocarriers, which coopted variable amounts of *GlyG* and constant amount of DSG, a slight decrease of the encapsulation efficiency was found, but this modification can be considered insignificant, considering the rather high values of the determined *EE*%, i.e., between 80.9 and 84% DSG, and 83.6 and 87.6% *GlyG* (for NLC-I systems prepared with plant oil from *Oenothera biennis*). Similar behavior was detected for NLC-II (with one exception) that showed a good ability to capture both categories of plant principles, hydrophilic and lipophilic (EE% determined were of 80 and 83.5% for DSG and between 81.2 and 89.5% for *GlyG*).

### 3.4. In Vitro Determination of Antioxidant Activity of NLC Based on Different Plant Active Principles

The TEAC method has been firstly used for the screening of antioxidant activity, being a method applicable for both, lipophilic and hydrophilic antioxidants. The ability to inhibit the stable free radical ABTS^•+^ by and loaded NLC solutions with individual or mixed plant extracts (DSG and *GlyG*), by comparison with the ABTS^•+^ elimination activity manifested by the native solutions of plant oils and phytochemicals is summarized in Figure 7. The behavior evaluation of the nano systems vs. solutions of the same concentration of plant active principles suggests a direct proportional relationship between encapsulated *GlyG* and ABTS^•+^ inhibitory capacity, also cumulated with a rather pronounced influence of the type of soybean and *Oenothera biennis* plant oils.

The antioxidant activity of native *GlyG* (up to 36.5% capacity to remove dangerous cation radicals) can be explained based on its chemical composition. *GlyG* has, in addition to glycyrrhizic acid (present in *GlyG* in the amount of 10%), other phenolic constituents such as flavonoids and isoflavonoids [41]. Although glycyrrhizic acid has extremely poor antiradical properties in the 0.1–100 μM concentration range [42], the presence of flavones and other phenolic compounds in *Glycyrrhiza glabra* may be responsible for the remarkable antioxidant properties of *GlyG*. The capturing of this phytochemical mixture into lipid nanocarriers resulted in an increase in antioxidant capacity. For instance, for NLC-I/II-*GlyG* was quantified an ability to inhibit up to 60.0% radical ABTS^•+^(for NLC-II-*GlyG*) and up to 64.2% (for NLC-I-*GlyG*).

On the opposite side are the NLC-I/II-DSG systems which show a weak ability to annihilate long-life radicals, the values being quite modest at ~13.4%. Despite the modest antioxidant properties of DSG, better results were obtained for dual-NLC systems that coopted *GlyG* and DSG. Coupling the two bio-active phytochemicals in the same nanostructured system and bringing them to the nanoscale has obviously led to an amplification of this property, as compared with free-NLC-I/II. Obvious amplification occurred in the case of NLC-II-DSG-*Gly* 4 which entrapped 0.5% DSG and 2% *GlyG*, for which an ABTS^•+^ radical inhibition capacity of 63.4% was determined (Figure 7).

A result that goes beyond the expectations line has been identified when comparing NLC-individual and NLC-dual systems, for example, NLC-I-*GlyG* (inhibition of 62.5%) and NLC-I-DSG-*GlyG* 3 (54%). Although the six dual NLC-systems (with both phytochemicals) show significant increases in inhibition rates compared to the free nanocarriers, NLC-I/II, only the dual NLC-I-DSG-*GlyG* 4 system achieves inhibition values comparable to those of NLC-individual ones/NLC-I/II-*GlyG.* These results may suggest the appearance of some restricted phenomena of the reaction between antioxidant *GlyG* with ABTS^•+^ radical, which could result in a decrease of the ability to transfer a hydrogen atom (from polyphenolic structures present in *GlyG*) to free radicals of ABTS^•+^.

For in vitro testing of the ability to capture short-lived oxygen radicals by the antioxidants present in NLC systems, chemiluminescence assay was used; this method is able to evaluate in particular the extinguishing capacity of reactive oxygen species (ROS) derived from hydrogen peroxide. A first important reference that emerges from the comparison of the results is the fact that both DSG and *GlyG* are effective against oxygen free radicals generated in the chemiluminescence system, compared to the situation presented above (that of long life ABTS^•+^ cation radicals).

Although encapsulation of DSG in NLC-I/II produces a moderate antioxidant effect (e.g., a ROS radical scavenger capacity between 54.7 and 56.3%), the association of DSG with *GlyG* leads to significantly superior antioxidant results (Figure 8). Obvious differences in free radical scavenging percentages were recorded between free NLC-I/II (up to 59% ability to scavenge the oxygen free radicals) versus NLC-*GlyG* and NLC-DSG-*GlyG*, whose values fall within the range 91 and 96% (with one exception, reported in the case of NLC-I/II-DSG-*GlyG* 1). In this case, the type of plant oil did not influence the NLC’s ability to capture ROS radicals. This amplified antioxidant activity, manifested on ROS in the chemiluminescent system, can be associated with: (i) the synergistic effect produced by the complex structures of the main bioactive compounds from the two phytochemicals, DSG and *GlyG*, but also the rich fatty acid composition of plant oils which may impact the results obtained; (ii) the nanosized effect obtained by coopting both lipophilic and hydrophilic extracts in the same distribution system, nanostructures that lead to the generation of many reaction centers for capturing free radicals.

### 3.5. Release Behavior of DSG and Glycyrrhizic Acid from Individual and Dual-NLC

Aqueous dispersions of individual-NLC (encapsulating DSG or *GlyG*) and dual-NLC (NLC-I/II-DSG-*GlyG*), were subjected to in vitro release studies using the dialysis method. With respect to the individual-NLC, the release of *GlyG* from the NLC-I and -II was similar (Figure 9a), not being influenced by the type of vegetable oil. The results were predictable given the previous results and the hydrophilic nature and affinity of the *GlyG* for the surfactant coating. As can be seen in Figure 9, in the first hour of the release study~19% Glycyrrhizin was determined for both NLC-I/II-*GlyG*, and a total release of *GlyG* occurred after 5 h (Figure 9a). A potential explanation for this behavior can be associated with the preferential distribution of *GlyG* at the NLC surface where, due to its strong hydrophilic character, it forms weak bonds (hydrogen bonds) with surfactants that envelop the lipid core, and as result, the dissolution of GlyG in the receiving environment occur very fast.

Although the receptor medium favored the relatively rapid release of both bioactive phytochemicals, in the case of NLC-I/II-DSG a slightly different behavior was detected, in the sense that in the first 5 h of the release study, DSG was released in proportions of 50% compared to 100% *GlyG* (detected for NLC-I/II-*GlyG* systems). In the first hour, a release of 4.3% DSG was found for the NLC-II-DSG system and 5.2% in the case of the NLC-I-DSG system. The slower release of DSG from NLC-I/II is directly influenced by DSG entrapment into nanocompartments in the core of lipid nanospheres (as can be seen from TEM micrographs). The exit from these nanocompartments and the traversing of the complex networks formed by the hydrocarbon chains in the lipid structure made the dissolution rate in this case to be significantly diminished and, implicitly, DSG showed a slower release. The lipophilic character of DSG and the low solubility in the receptor medium are also factors determined in the behavior of its release from NLC.

It is worth noting that for the dual-NLC systems, there is a slower release of both lipophilic and hydrophilic bioactive components, more pronounced in the case of hydrophilic *GlyG*, compared to NLC loaded with only one of the two phytochemicals. For instance, individual-NLC released ~81% *GlyG* after 4 h of experiments, while 39% *GlyG* was released from dual-NLC (from NLC-II-DSG-*GlyG*, Figure 9c), and 61% (from NLC-I-DSG-*GlyG*, Figure 9b).

Concerning the release rate of the lipophilic DSG, the release trend is similar for both NLC-I and II, after 4 h of release being determined relatively equal amounts of DSG in the receptor environment, e.g., about ~30% DSG. This uniformity of DSG release can be attributed to the lipid core formed by the solid lipids with *Glicyne max* and *Oenothera biennis* oils. In contrast, *GlyG* release from NLC was significantly delayed in the case of NLC-II (Figure 9b) compared to NLC-I (Figure 9c).

### 3.6. In Vitro Determination of Cytotoxic Effect of Individual- and Dual Herbal Active Loaded-NLC (by MTS and RTCA Assays)

An MTS method was initially performed to measure the cytotoxic effect of developed-NLC. To evaluate the toxicological profile of each free-NLC and NLC loaded with herbal principles, normal HUVEC cell cultures were treated for 24 h and 48 h with scalar concentrations of NLC (between 200 and 3125 µg/mL). The results of the cytotoxicity study (e.g., the quantification of the number of viable cells) show that the viability of HUVEC endothelial cells is significantly influenced by the concentrations of NLC, which is affected when the cells are subjected to concentrations higher than 100 μg/mL NLC (Figure 10). A first treatment period of 24 h underlined that the cell viability was higher than 65% for concentrations lower than 50 mg/mL (with a few exceptions), which indicates a very low cytotoxicity effect induced by treatment with NLC-I/II-DSG-*GlyG* (Figure 10a).

Prolonged NLC treatment on HUVEC endothelial cells (48 h) led to a counterbalance of cell viability as compared to the previous results obtained after 24 h. Particularly, a further increase in the cell’s survivability was registered (Figure 10b). The occurrence of cell recovery/proliferation phenomena was found after 48h treatment, in this case determining endothelial cell viability values > 80%, in the range of concentrations 50–25 mg/mL (Figure 10b). According to the obtained results, the most efficient nanocarrier systems are the dual-NLC systems for which after treatment with 50 mg/mL the value of cell viability was 81.9% for NLC-II-DSG-*GlyG* 4). Based on the present observations, the NLCs were not cytotoxic toward endothelial normal cells.

Proliferative versus cytotoxicity capacity was complemented by a specific real-time assay (RTCA test) of free- and herbal-loaded NLC on HUVEC endothelial cells. RTCA measures the cell index at any time, indicating the treatment time of the NLC concentration that determines a viability/cytotoxicity of 50% (IC50). The following aspects were highlighted by RTCA results: (i.) Confirmation of the results obtained by MTS assay. At high concentrations, e.g., 400 and 200 µg/mL, cell viability decreases significantly, indicating the appearance of a cytotoxic effect manifested by NLC (Figure 11). It is interesting to note that by increasing treatment times, with a few exceptions, values comparable to those of untreated cells are reached, which indicates a lack of cytotoxic effects. (ii.) Detection of a safety NLC use at concentrations between 25 and 100 µg/mL, associated with increased cell viability (comparable to those of untreated cell control, represented by the red curve, Figure 11). (iii.) According to IC50 values (Table 3), a minimal cytotoxic effect was reached in the HUVEC cell line after treatment with developed herbal-NLC; as previously mentioned, prolonged treatment (48 h) leads to an increase in cell viability which confers a noticeable difference between IC50 values after 24 h and 48 h of treatament. For instance, acceptable IC50 values were recorded for nanocarriers prepared with soybean oil; a concentration of 159.92 ± 1.61 μg/mL NLC-II-DSG-*GlyG* 4 resulted in a 50% inhibition of cell growth (Table 3).

### 3.7. In Vitro Performance of Dual Herbal Actives-NLC on Anti-Inflammatory Action

Cytokines are peptides and proteins mainly secreted by activated immune cells, involved in the signaling between the cells of the immune system [43], their main function being to mediate and regulate the immune inflammatory response. TNF-α, IL-1, IL-6 and IL-12 are the main cytokine of macrophage phagocytosis, antigen presentation and inflammation regulation [20]. The detection of cytokines activity is of great significance due to its effects on almost all biological processes in the body and can be either harmful or beneficial, depending on the quantities produced and the conditions around production [44].

By expression evaluation of the pro-inflammatory cytokines TNF-α and IL-6 (ELISA assay), revealed a strong anti-inflammatory effect. By treating HUVEC cells with NLC, the production of pro-inflammatory cytokines TNF-alpha and IL-6 was significantly inhibited (Figure 12).

In terms of the dose applied to the treatment of endothelial cells, it was found that treatments with 200 µg/mL NLC led to a decrease in the levels of inflammatory markers TNF-α and IL-6, respectively to a lower degree of inhibition, compared with a dose of 50 µg/mL. For a dose of 50 µg/mL was registered a significant increase in the inhibition percent of both categories of proinflammatory cytokines released. This more effective counteracting of the lower dose of NLC, respectively 50 mg/mL, compared to the treatment with a higher concentration, 200 mg/mL, can be explained based on the previous results of MTS and RTCA analyzes that showed a decrease in cell viability at concentrations of 200 mg/mL. This effect implicitly leads to denaturation and cell death, which directly results in a percent decrease in inhibition of TNF-α and IL-6 cytokines.

By comparing the anti-inflammatory effect of individual- and dual herbal active-NLC on TNF-α cytokine release versus interleukin 6 levels, more effective inhibition of TNF-α release was detected. These results may be associated with the action of the oxidizing agent (H_2_O_2_) applied to HUVEC cells. The action of H_2_O_2_ resulted in a higher release of TNF-alpha compared to untreated cells compared to IL-6.

It should be noted that unloaded-NLC showed themselves a good ability to inhibit the release of pro-inflammatory cytokines, e.g., for concentrations of 50 mg/mL, inhibition percent of IL-6 was higher than 50%, and in case of TNF-α the inhibition exceeded 85% (Figure 12). These results suggest the obvious impact of *Oenothera biennis* and *Glicyne max* oil on the anti-inflammatory action. A comparative look at NLC-I versus NLC-I-DSG-*GlyG* shows a more pronounced degree of inhibition of NLC-I-DSG-*GlyG,* detected for TNF-α (e.g., 96.9% for dual loaded-NLC versus 89.8% for NLC-I). A slight amplification of anti-inflammatory action for IL-6 was also measured (e.g., IL-6 inhibition of 53.9% for NLC-I versus 62% detected for NLC-I-DSG-*GlyG*). In the case of NLC-II, they provided a comparable inhibition to NLC-I systems: 62.5% IL-6 inhibition was detected for NLC-II-DSG-*GlyG*, while for TNF-α the inhibition percentage was 97.9% (Figure 12).

## 4. Conclusions

The phytochemicals present in medicinal plants encapsulated by nanostructured carriers will play an increasing role in therapeutics in the future. Detailed and unique research was undertaken to understand the role and main contribution of phytochemicals for obtaining herbal enriched-nanostructured carriers with improved therapeutic efficacy. Coupling of selected phytochemicals from various herbal sources with lipid nanocarriers offers multiple opportunities, mainly to develop new formulations with better properties, to increase the effectiveness of herbals delivery and to improve the most desirable properties of herbal medicine, i.e., bioavailability, antioxidant, and anti-inflammatory properties.

Various lipid nanocarriers (prepared with oils from *Oenothera biennis* and from *Glicyne max*) were suitable for entrapping bioactive phytochemicals (Diosgenin and *Glycyrizia Galbra*). Spherical lipid nanocarriers (less than 200 nm) with nano spheric inclusions uniformly dispersed in the lipid core (<5 nm), were detected by phase contrast micrography. The preferential affinities for the lipophilic lipid core (case of DSG) or for the coating made by the surfactants (case of *Glycyrrhiza glabra* extract) was underlined by differential scanning calorimetry results. The chromatographic results highlighted the remarkable ability of lipid nanocarriers to capture both categories of phytochemicals with different polarities; encapsulation efficiency was found to be higher than 80% for DSG, and almost 90% for *GlyG,* respectively. Safety herbal-lipid nanocarriers were confirmed by MTS and RTCA assays. According to the in vitro results, the dual herbal actives-NLC were not cytotoxic toward HUVEC endothelial normal cells at concentrations between 25 and 100 µg/mL.

The in vitro release profile revealed that dual herbal-based nanoparticles allow sustained release which further ensures phytochemicals to reach inflamed cells, where DSG and *GlyG* have their therapeutic target. Dual herbal actives-NLC showed an elevated ability to annihilate both kinds of long-live cationic radicals (ABTS^•+^) and short life oxygenated radicals (ROS). For lipid nanocarriers that coopted both kinds of lipophilic and hydrophilic herbal principles an inhibition capacity of 63.4% ABTS^•+^ was determined, while the ability to capture ROS radicals falls within the range between 91 and 96%. Moreover, the production of pro-inflammatory cytokines TNF-alpha and IL-6 was significantly inhibited by the lipid nanocarriers that coopted both kinds of lipophilic and hydrophilic herbal principles. By comparing the anti-inflammatory action of dual herbal actives-NLC, a more effective inhibition of TNF-α release was detected than for IL-6 interleukin.

These results provide a real and comprehensive basis for the use of dual herbal-NLC in pharmacotherapy, with potential application in the treatment of inflammatory diseases. There has been progress on herbal drugs and nanocarriers, indicating that lipid nanocarriers and herbal medicines will be integrated and applied in the future.

## Figures and Tables

**Figure 1 pharmaceutics-13-01511-f001:**
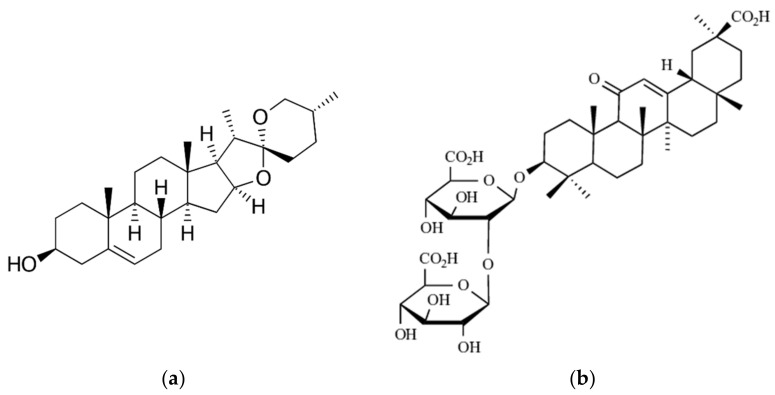
Bio-active phytocompounds from plant extracts: (**a**) Diosgenin from wild yam extract/*Discorea villosa*; (**b**) Glycyrrhizic acid from licorice extract/*Glycyrrhiza glabra*.

**Figure 2 pharmaceutics-13-01511-f002:**
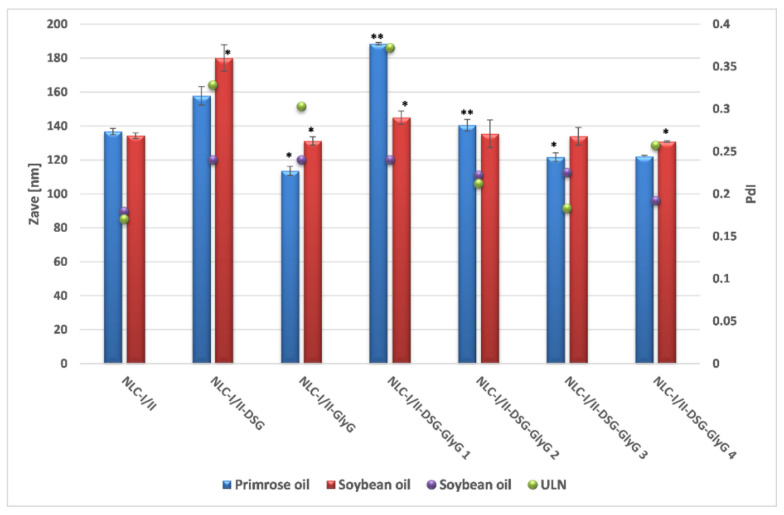
Variation in size and polydispersity index (PdI) depending on the concentration and type of encapsulated plant principle (DSG and *GlyG*).* *p* < 0.05; ** *p* < 0.005; NS *p* < 0.01. Data are expressed as mean ± SD, *n* = 3 NLCI/II vs. other groups.

**Figure 3 pharmaceutics-13-01511-f003:**
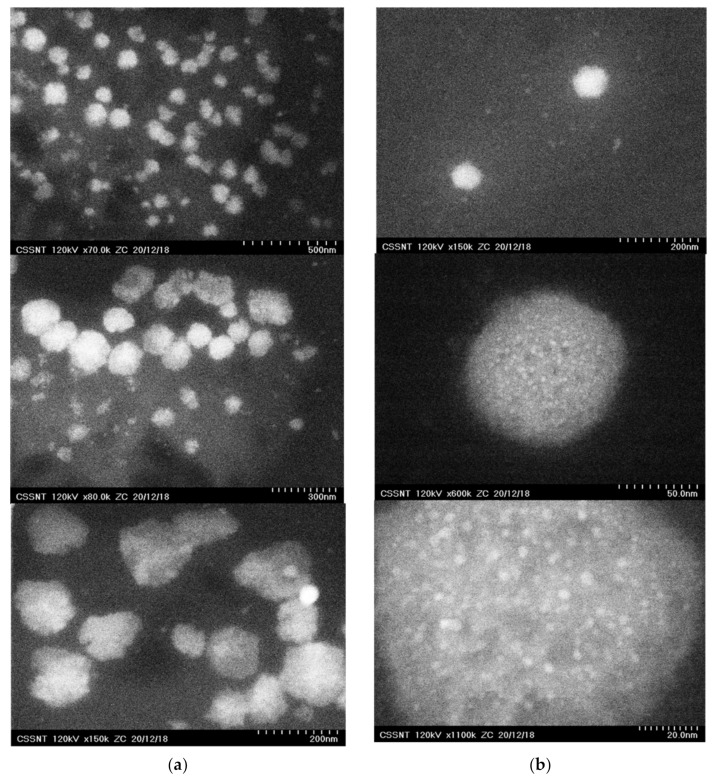
TEM images of (**a**) NLC-II-DSG-*GlyG* 4 (obtained by using soybean oil from *Glicyne max*) and (**b**) NLC-I-DSG-*GlyG* 4 (obtained by using plant oil form *Oenothera biennis*).

**Figure 4 pharmaceutics-13-01511-f004:**
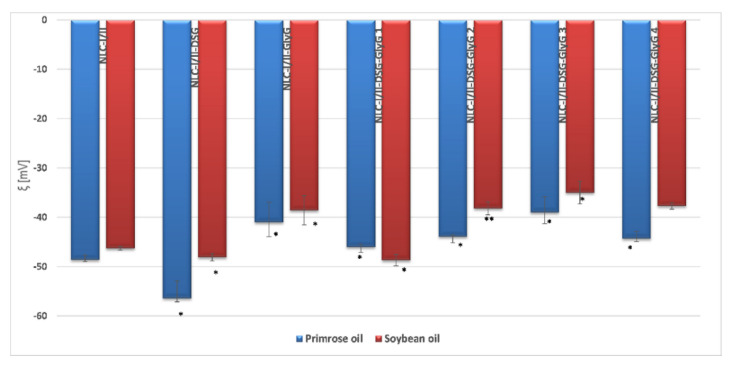
Variation of electrokinetic potential depending on the concentration of active ingredients (DSG and *GlyG*) and the type of plant oil used in the preparation of NLC. * *p* < 0.05; ** *p* < 0.005; NS *p* < 0.01. Data are expressed as mean ± SD, *n* = 3 NLCI/II vs. other groups.

**Figure 5 pharmaceutics-13-01511-f005:**
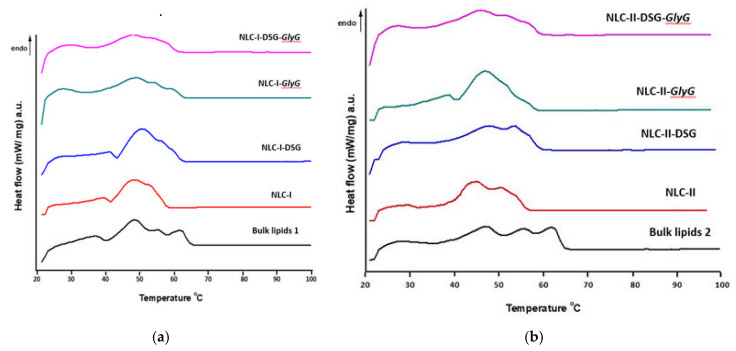
DSC curves of individual- and dual-NLC vs. NLC without phytochemicals: (**a**) the set of NLC-I formulations and (**b**) the set of NLC-II formulations.

**Figure 6 pharmaceutics-13-01511-f006:**
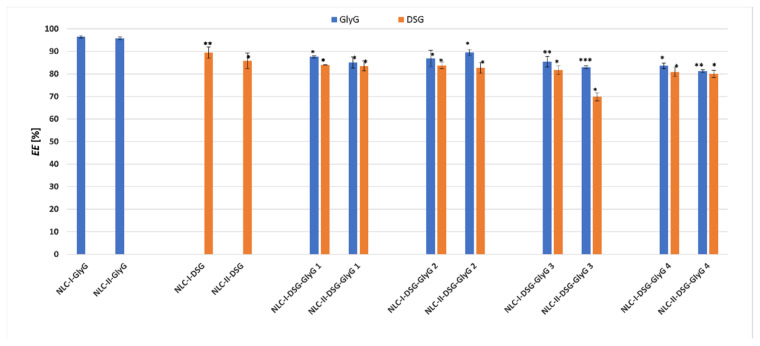
Encapsulation efficiency (*EE*%) of DSG and glycyrrhizic acid (from *GlyG*) entrapped into NLC-I and II. * *p* < 0.05; ** *p* < 0.005; *** *p* < 0.0005; NS *p* < 0.01. Data are expressed as mean ± SD, *n* = 3 NLCI/II-GlyG/DSG vs. other groups.

**Figure 7 pharmaceutics-13-01511-f007:**
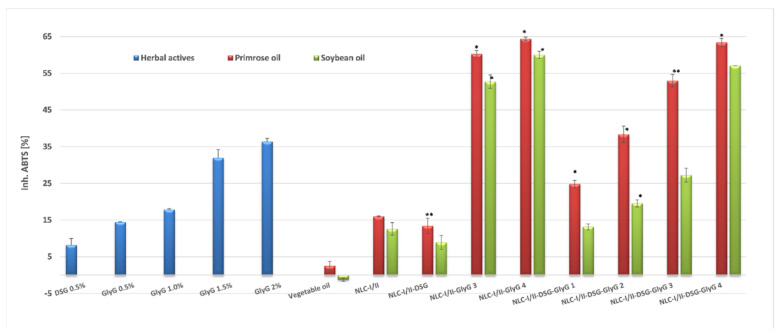
Variation of ABTS^•+^ inhibition capacity by NLC systems loaded with a single plant active principle (GlyG or DSG) versus NLC that entrap two plant bioactive principles (GlyG and DSG). * NLC-I/II-*GlyG* 3 and 4 are lipid nanocarriers loaded only with *GlyG* (1.5% and 2% *GlyG*). * *p* < 0.05; ** *p* < 0.005; NS *p* < 0.01. Data are expressed as mean ± SD, *n* = 3 NLCI/II vs. other groups.

**Figure 8 pharmaceutics-13-01511-f008:**
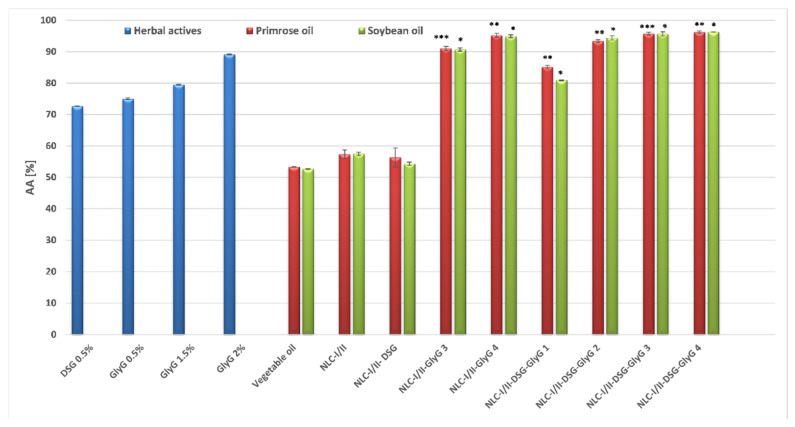
In vitro results of antioxidant activity of free and NLC loaded with phytochemicals, by chemiluminescence assay. * *p* < 0.05; ** *p* < 0.005; *** *p* < 0.0005; NS *p* < 0.01. Data are expressed as mean ± SD, *n* = 3 NLCI/II vs. other groups.

**Figure 9 pharmaceutics-13-01511-f009:**
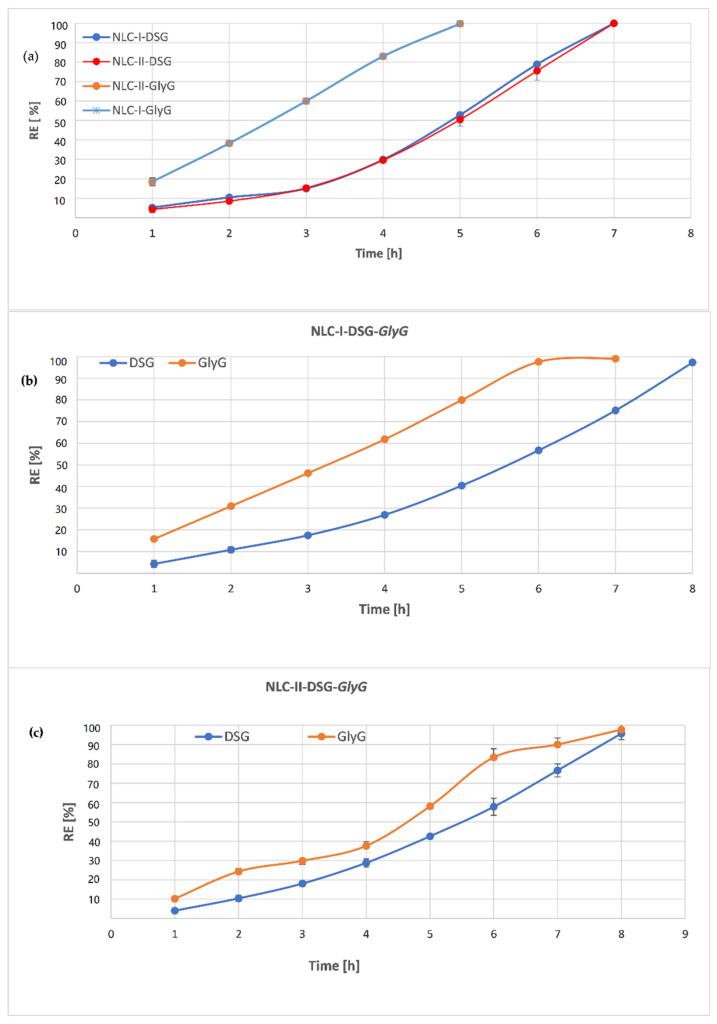
In vitro release profile of herbal principles from: (**a**). Individual-lipid nanocarriers (NLC-I/II-DSG si NLC-I/II-*GlyG*); (**b**,**c**). Dual-lipid nanocarriers (NLC-I/II-DSG-*GlyG* 4).

**Figure 10 pharmaceutics-13-01511-f010:**
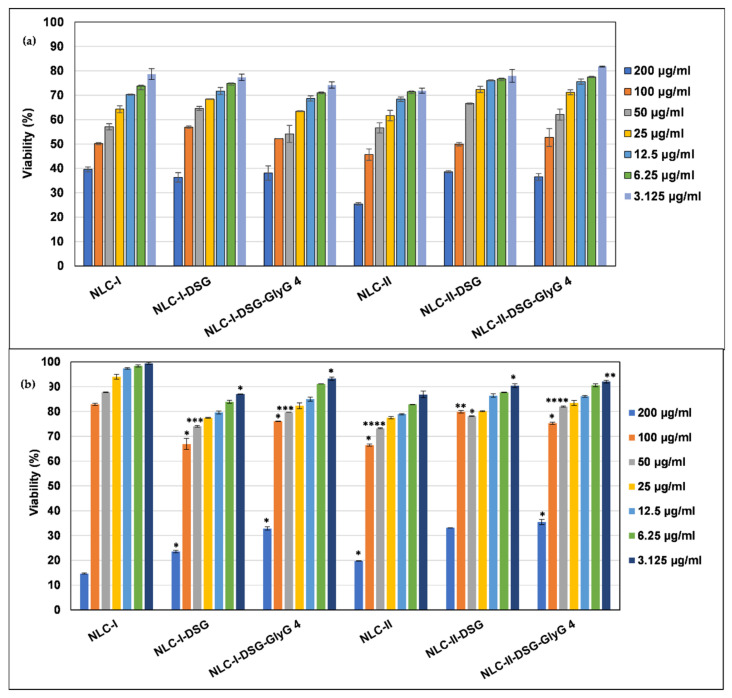
The effect of individual- and dual herbal actives loaded-NLC on cell viability of HUVEC endothelial cells: (**a**). after 24 h f treatment; (**b**). after 48 h of treatment. * *p* < 0.05; ** *p* < 0.005; *** *p* < 0.0005; **** *p* < 0.00005; NS *p* < 0.01. Data are expressed as mean ± SD, *n* = 3 NLCI/II vs. other groups.

**Figure 11 pharmaceutics-13-01511-f011:**
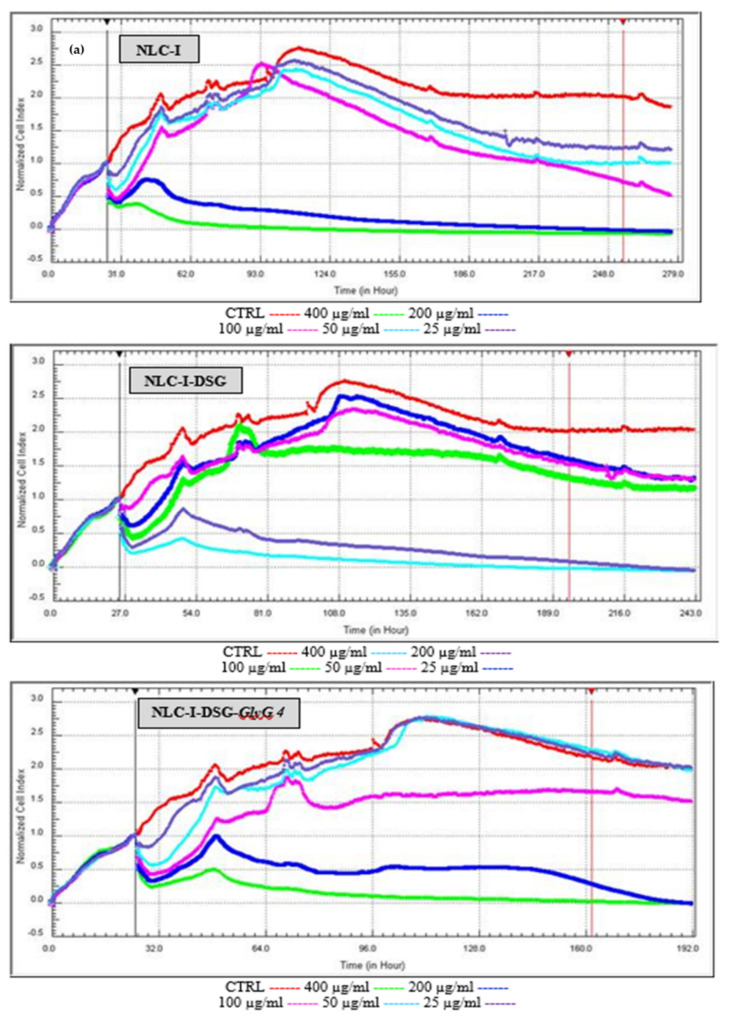
Cytotoxic vs. proliferation action induced by free and loaded NLC-I/II on normal endothelial cells, by RTCA assay (**a**). NLC-I = lipid nanocarriers prepared by using primrose oil; (**b**). NLC-II = lipid nanocarriers prepared with soybean oil).

**Figure 12 pharmaceutics-13-01511-f012:**
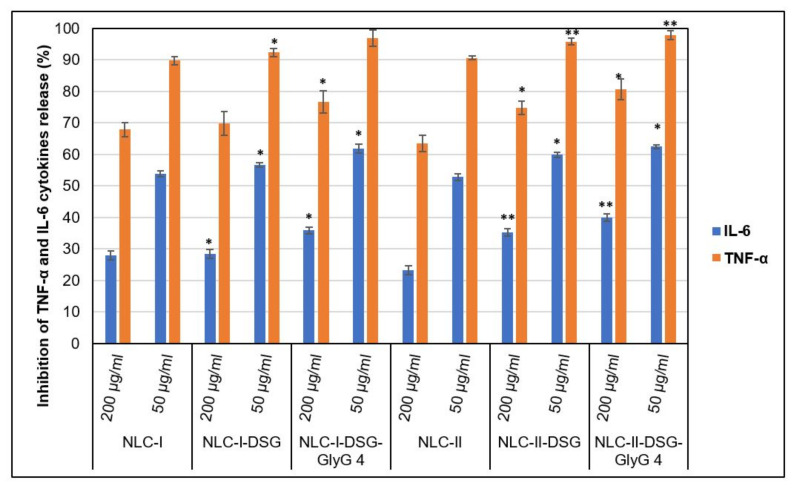
Inhibition action of dual herbal actives-NLC on the release of TNF-α and IL-6 cytokines. * *p* < 0.05; ** *p* < 0.005; NS *p* < 0.01. Data are expressed as mean ± SD, *n* = 3 NLCI/II vs. other groups.

**Table 1 pharmaceutics-13-01511-t001:** The composition of the individual- and dual herbal-NLC.

NLC Formulation	Lipid Phase, 10%			Surfactant Mixture, 2.5%			DSG %	*GlyG* %
	CP	GMS	Veg. Oil *	Co-Polymer	Polysorbate 20	Phosphatidyl Choline		
NLC-I/II *	3.5	3.5	3	0.375	1.75	0.375	-	-
NLC-I/II-DSG							0.5	-
NLC-I/II-GlyG **							-	2
NLC-I/II-DSG-*GlyG* 1							0.5	0.5
NLC-I/II-DSG-*GlyG* 2							0.5	1
NLC-I/II-DSG-*GlyG* 3							0.5	1.5
NLC-I/II-DSG-*GlyG* 4							0.5	2

* NLC-I and NLC-II are lipid nanocarriers prepared by using different vegetable oil, e.g., primrose oil from *Oenothera biennis* (NLC-I) and soybean oil from *Glicyne max* (NLC-II).** NLC-I/II-*GlyG* 1, 2, 3 and 4 are lipid nanocarriers loaded only with *GlyG* (0.5%, 1%, 1.5% and 2% *GlyG*).

**Table 2 pharmaceutics-13-01511-t002:** DSC characteristics for NLC-I/II-DSG-*GlyG*.

NLC Sistem	Tm (°C)	ΔHm (J/g)	CI (%)
NLC-I	40.2	4.42	
	50.8	52.25	28.30
NLC-I-DSG	38.9	6.31	
	47.7	70.91	38.41
NLC-I-*GlyG*	47.7	60.59	32.82
NLC-I-DSG-*GlyG* 4	47.1	54.87	29.72
Physical mixture of I	37.4	7.61	100
	48.6	18.46	
	55.9	1.84	
	61.7	9.01	
NLC-II	43.4	45.35	34.64
	50.5	7.96	
NLC-II-DSG	48.1	57.55	
	53.8		43.96
NLC-II-*GlyG*	48.8	47.21	36.07
NLC-II-DSG-*GlyG* 4	47.1	57.63	44.03
Physical mixture of II	47.1	13.09	100
	55.5	3.8	
	62	9.51	

**Table 3 pharmaceutics-13-01511-t003:** IC50 values for individual- and dual herbal active-loaded NLC.

Herbal Actives Loaded-NLC	IC50 * (µg/mL) 24 h	IC50 (µg/mL) 48 h
NLC-I	91.44 ± 2.05	137.53 ± 1.40
NLC-I-DSG	127.05 ± 1.13	127.79 ± 1.68
NLC-I-DSG-*GlyG* 4	96.91 ± 1.06	152.82 ± 2.14
NLC-II	79.74 ± 2.18	119.19 ± 1.51
NLC-II-DSG	101.17 ± 1.27	163.19 ± 2.40
NLC-II-DSG-*GlyG* 4	102.00 ± 1.26	159.92 ± 1.61

* IC50 is presented as mean + SD of three independent experiments.

## Data Availability

Not applicable.

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
