# Peer review of "Multifaced Role of Dual Herbal Principles Loaded-Lipid Nanocarriers in Providing High Therapeutic Efficacity"

_pharmaceutics, 2021, doi:10.3390/pharmaceutics13091511_

Round 1

Reviewer 1 Report

Lacatusu et al studied the multifaced role of dual herbal principles loaded-lipid nanocarriers in providing high therapeutic efficacity. The train of thought of the research is clear, which provides a reference for the later research. However, there are some deficiencies, which need to provide more evidence and modifications.

Q1: Statistical analysis was conducted in this paper. Please mark the statistical differences in the relevant figures.

Q2: Abstract is separate from the text, please replace the abbreviations with full names. In addition, in the line 120, Glycyrrhiza glabra should be abbreviated as GlyG. Please check and modify the abbreviations in this paper.

Q3: It is better to incorporate diosgenin as the keyword.

Q4: In “2.1 Materials”, Glycyrrhiza glabra extract was standardized in 10% glycirrhizic acid/glycyrizin, please provide the detail chromatogram or other solid evidence.

Q5: Please specify NLC-I/II-GlyG 3 and NLC-I/II-GlyG 4 in Figure 7 and NLC-0.5% EYS in Figure 8. Besides, Figure 11 is not clear enough.

Q6: As the authors stated, “At high concentrations, eg 400 and 200 μg/ml, cell viability decreases significantly, indicating the appearance of a cytotoxic effect manifested by NLC.” Why choose 200 μg/ml in the in vitro anti-inflammatory assay?

Q7: In my opinion, cytotoxic effect and antioxidant activity of NLC should be expressed as IC50.

Q8: Please check the references. For example, ref.5 has nothing to do with “The World Health Organization reported that over 80% of the world's population relies on herbal medicines (phytomedicine) for a particular aspect of their primary health care needs [4,5].”

Author Response

Dear Reviewer,

Based on the helpful Reviewer comments, we have carefully reviewed the manuscript.

Thank you for the valuable comments and suggestions which help us in further improving of the manuscript “Multifaced role of dual herbal principles loaded-lipid nanocarriers in providing high therapeutic efficacity”.

All suggestions and observations provided by the Reviewer have been addressed and presented in the revised manuscript with a green  background.

Best regards,

Prof. Nicoleta Badea

Reviewer 2 Report

The paper is focused on the multifaced role of dual herbal principles loaded-lipid nanocarriers for therapeutic purposes. Specifically, two herbal principles, namely diosgenin, DSG and Glycyrrhiza glabra extract, GlyG) coopted in the Nanostructured Lipid Carriers (NLC) were developed for improving the antioxidant and anti-inflammatory actions. The overall contribution of phytochemicals, vegetable oils and of lipid matrices was highlighted by a comparative study of size, stability, entrapment efficiency, morphological characteristics, and thermal behavior. The study may open new pharmacotherapy horizon, it provides a comprehensive basis for the use of herbals-NLC in the treatment of inflammatory diseases.

The work was conducted with adequate means A critical comparison with previous works on the same field was also reported. I suggest some revision prior to eventual publication.

  • English needs an extensive revision. Many misspellings can be found throughout the whole text e.g. title: therapeutical efficacy; abstract, line 7: behavior, etc.
  • This section needs to be shortened highlighting the main outcomes of the work and potential future applications.

Author Response

Dear Reviewer,

Based on the helpful your comments, we have carefully reviewed the manuscript.

Thank you for the valuable comments and suggestions which help us in further improving of the manuscript “Multifaced role of dual herbal principles loaded-lipid nanocarriers in providing high therapeutic efficacity”.

 All suggestions and observations provided by the Reviewers have been addressed and presented in the revised manuscript with a green  background.

Best regards,

Prof. Nicoleta Badea

Round 2

Reviewer 1 Report

 Only one suggestion: It is better to introduce NLC-I/II-GlyG 3 and NLC-I/II-GlyG 4 in "2. Materials and Methods".

Author Response

Dear Reviewer,

Based on the helpful Reviewer comments, we have carefully reviewed the manuscript.

Thank you for the valuable comments and suggestions which help us in further improving of the manuscript “Multifaced role of dual herbal principles loaded-lipid nanocarriers in providing high therapeutic efficacity”.

 The suggestion provided by the Reviewer has been addressed and presented in the revised manuscript with a yellow  background.

Best regards,

Prof. dr. Nicoleta Badea
